# Emerging Mosquito-Borne Threats and the Response from European and Eastern Mediterranean Countries

**DOI:** 10.3390/ijerph15122775

**Published:** 2018-12-07

**Authors:** Nicholas Johnson, Mar Fernández de Marco, Armando Giovannini, Carla Ippoliti, Maria Luisa Danzetta, Gili Svartz, Oran Erster, Martin H. Groschup, Ute Ziegler, Ali Mirazimi, Vanessa Monteil, Cecile Beck, Gaelle Gonzalez, Sylvie Lecollinet, Houssam Attoui, Sara Moutailler

**Affiliations:** 1Animal and Plant Health Agency (APHA), Woodham Lane, Addelstone, Surrey KT15 3NB, UK; Mar.Fernandez@apha.gsi.gov.uk; 2Faculty of Health and Medicine, University of Surrey, Guildford, Surrey GU2 7XH, UK; 3Istituto Zooprofilattico Sperimentale dell’Abruzzo e del Molise Guiseppe Capporale, Campo Boario, 64100 Teramo, Italy; a.giovannini@izs.it (A.G.); c.ippoliti@izs.it (C.I.); m.danzetta@izs.it (M.L.D.); 4Division of Virology, Kimron Veterinary Institute, POB 12, Bet Dagan 50250, Israel; giliun@gmail.com (G.S.); orane@moag.gov.il (O.E.); 5Friedrich-Loeffler-Insitut, Südufer 10, 17493 Greifswald, Insel Riems, Germany; martin.groschup@fli.de (M.H.G.); ute.ziegler@fli.de (U.Z.); 6Karolinska Institute, Nobel vägsalle, 16161 Stockholm, Sweden; ali.mirazimi@ki.se (A.M.); vanessa.monteil@ki.se (V.M.); 7UMR 1161 Virologie, Animal Health Laboratory, ANSES, INRA, Ecole Nationale Vétérinaire d’Alfort, Université Paris-Est, 94700 Maisons-Alfort, France; cecile.beck@anses.fr (C.B.); gaelle.gonzalez@anses.fr (G.G.); sylvie.lecollinet@anses.fr (S.L.); houssam.attoui@vet-alfort.fr (H.A.); 8UMR BIPAR, Animal Health Laboratory, ANSES, INRA, Ecole Nationale Vétérinaire d’Alfort, Université Paris-Est, 94700 Maisons-Alfort, France; sara.moutailler@anses.fr

**Keywords:** West Nile virus, Usutu virus, mosquito, *Culex pipiens*, *Aedes albopictus*

## Abstract

Mosquito-borne viruses are the cause of some of the greatest burdens to human health worldwide, particularly in tropical regions where both human populations and mosquito numbers are abundant. Due to a combination of anthropogenic change, including the effects on global climate and wildlife migration there is strong evidence that temperate regions are undergoing repeated introduction of mosquito-borne viruses and the re-emergence of viruses that previously were not detected by surveillance. In Europe, the repeated introductions of West Nile and Usutu viruses have been associated with bird migration from Africa, whereas the autochthonous transmission of chikungunya and dengue viruses has been driven by a combination of invasive mosquitoes and rapid transcontinental travel by infected humans. In addition to an increasing number of humans at risk, livestock and wildlife, are also at risk of infection and disease. This in turn can affect international trade and species diversity, respectively. Addressing these challenges requires a range of responses both at national and international level. Increasing the understanding of mosquito-borne transmission of viruses and the development of rapid detection methods and appropriate therapeutics (vaccines / antivirals) all form part of this response. The aim of this review is to consider the range of mosquito-borne viruses that threaten public health in Europe and the eastern Mediterranean, and the national response of a number of countries facing different levels of threat.

## 1. Introduction

Europe is dominated by a temperate climate although there is great variation affected by proximity to seas and elevation. Simplistically these range from a Mediterranean climate (warm and dry) in the south, to oceanic in the west (cool summers, mild winters), continental (cold winters and warm summers) to polar in the far north. Despite this variation, mosquitoes are present across the whole continent although the species assemblage and seasonal abundance varies dramatically. A consistent phenomenon is that the summer months in Europe (July to September) support mosquito activity whereas the winter months (December to February) generally lead to a cessation of activity. Ecological studies in southern [1] and northern [2] Europe illustrate this with both showing peak abundance of particular species during the summer months. It is during these peaks that autochthonous virus transmission takes place and when outbreaks of mosquito-borne disease are reported.

In addition to native mosquitoes, Europe has been colonized by a series of invasive aedine mosquitoes (Table 1) that have been reviewed extensively [3,4]. The initial introduction of each, and subsequent spread have all resulted from human translocation of desiccated eggs, for example in used tires, or movement of adults in cars [5]. The rapid spread of these species is illustrated by the introduction of *Ae. albopictus* into Spain. The mosquito was first detected in 2004 in the city of Sant Cugat del Vallès in the northeast of the country [6]. Within ten years extensive surveillance has detected the species along the entire eastern coastline of Spain and evidence of spread to municipalities inland [7]. This expansion reflects the distribution of *Ae. aegypti*, a species last reported in Spain in 1939 [8]. In addition to being an aggressive anthropophilic feeder and thus a nuisance, it is also a vector for a growing number of viruses, so the presence of abundant populations of *Ae. albopictus* raises the risk of autochthonous disease transmission within the human population wherever it is found. This problem is affecting an increasing number of countries around Europe.

Autochthonous transmission of viruses by mosquitoes in Europe has been dominated by the emergence of a series of exotic viruses. The best example of this has been the repeated introduction of West Nile virus (WNV), transmitted predominantly by mosquito species of the genus *Culex* [9]. Italy in particular has reported annual outbreaks of West Nile fever (WNF) in both human and equine populations caused by lineage 1 WNV [10]. A second lineage (lineage 2) has been responsible for outbreaks in Hungary and Greece [11]. In 2018, as of the 27th September, European Union (EU) Member States reported 1,266 human cases: Italy (n = 495), Greece (n = 281), Romania (n = 237), Hungary (n = 190), Croatia (n = 44), France (n = 18), Austria (n = 15), Bulgaria (n = 5) and Slovenia (n = 3). European Union neighbouring countries reported 404 human cases: Serbia (n = 320), Israel (n = 81) and Kosovo (n = 3). A total of 124 deaths due to WNF have been reported [12]. Yearly totals of autochthonous human cases in the EU between 2011 to 2017 range from 74 cases in 2014 to 242 cases in 2012 [13]. The observed trend (Figure 1) suggests that in 2018, when only half of the transmission season has elapsed, the number of recorded human cases is already higher than any previous year on record. The epidemiology of WNV is complex, consisting of an infection cycle between mosquitoes species vectoring disease and wild birds acting as the reservoir, but with spill over to humans and horses that can result in disease. This is driven by bridge vectors, mosquito populations that bite both avian and mammalian species. Examples of this include populations of *Culex pipiens* [14] especially those in southern Europe, and *Culex modestus* [15].

The presence of *Ae. albopictus* in southern Europe has provided fertile ground for the introduction of Aedes mosquito-transmitted viruses by viraemic humans travelling from endemic regions. Examples of this include outbreaks of fever caused by dengue virus (DENV) in southern France [16] and chikungunya virus (CHIKV) in Italy [17], and has led to concerns that Zika virus (ZIKV) could also be introduced in a similar manner. Within avian species, the introduction of Usutu virus (USUV) into southern Europe has caused a progressive expansion of the area affected [18], leading to a marked increase in mass mortality of species such as the common blackbird (*Turdus merula*). Arguably this is a significant threat to avian biodiversity in many regions of Europe [19].

The variety of mosquito-borne viral threats (Table 2), their seasonality and sporadic occurrence, presents a distinct challenge to devising a surveillance strategy as many options are available from surveillance for infection in the vector, reservoir species and spill-over hosts. A balance needs to be struck between defining the desired outcome, for example providing warning of infection in humans before it occurs, and budgetary limitations that mean not all surveillance options can be pursued. In addition, the use of multiple animal sources often requires a range of specialties (entomologists, veterinarians, wildlife biologists) that demands integration of data from different organisations [20]. This is often described as a One Health strategy [21]. The purpose of this review is to consider the challenges faced by a number of countries within and close to Europe to the threats posed by mosquito-borne disease outbreaks.

## 2. National Mosquito-Borne Threats

### 2.1. United Kingdom

The UK is free of pathogenic mosquito-borne viruses and the only recorded arthropod-borne virus is louping ill virus, a pathogen in sheep and grouse transmitted by ticks [22]. Despite this absence, there is evidence that the indigenous mosquito species are capable of transmitting exotic arboviruses including Japanese encephalitis virus [23], WNV [24], Rift Valley fever virus [25] and USUV [26]. Serology studies suggested that WNV, USUV and Sindbis virus (SINV) were circulating in wild bird populations [27] and chicken flocks [28] but subsequent surveillance studies have not confirmed these observations [29,30,31]. However, the proximity of USUV affecting wild bird populations in northwestern Europe [32] has heightened the risk of the introduction of this virus in recent years. In addition, a bridge mosquito vector for flaviviruses, *Culex modestus*, has been detected in areas adjoining the Thames Estuary in southeast England [33] and its distribution appears to be increasing [34]. Despite the presence of WNV in southern Europe for a similar time to USUV, it has not been detected in regions adjoining the English Channel.

Surveillance for arthropod-borne disease is based on detection of human cases by Public Health England (PHE). However, these infections result from travel to disease-endemic areas, particularly ZIKV following its introduction into South America [35], rather than autochthonous transmission. Reports of travel related cases of WNV in northern Europe are surprisingly infrequent [36,37], perhaps related to the low number of infected individuals who develop severe fever and the sporadic nature of outbreaks in Europe. Surveillance in animals, conducted by the Animal and Plant Health Agency (APHA), is limited to passive sampling for WNV in wild birds between April and October [20]. Detection of virus in equids is limited to syndromic surveillance where disease is suspected. So far this has detected a single case of WNV infection in a horse imported from Cyprus [38]. Focused surveys of mosquitoes in southern England have failed to detect WNV in pooled samples [31]. Mosquito surveillance at UK ports of entry is aimed at detecting incursions by invasive mosquito species [39]. In recent years this has detected *Ae. albopictus* eggs in the south of England [40], although there is no evidence that the species has established a resident population. This level of surveillance is considered proportionate to the risk of introduction. Currently, the risk of WNV introduction into the UK is considered very low [41].

### 2.2. Sweden

Two mosquito-borne viruses have been detected in Sweden [42]. The clinically more important SINV, the cause of polyarthritis, fever and skin rash, causes sporadic outbreaks in Sweden from transmission by a range of mosquito species [43]. Inkoo virus causes a mild febrile illness or is asymptomatic and has also been isolated from Scandinavian mosquitoes [44]. Both viruses have been detected in mosquito larvae suggesting that transovarial transmission may be a mechanism for virus persistence between summers [45]. There has been no evidence of WNV or USUV transmission within Sweden or other Scandinavian countries.

### 2.3. France

France faces the circulation of two mosquito-borne-flaviviruses with episodic re-emergence of WNV over a long period and the first introduction of USUV in 2015. The first reported WNV outbreak that affected humans and horses took place in the southeast of France in the 1960s, in the Camargue region, an area of natural wetland on the Mediterranean coast between the cities of Montpellier and Marseille. The re-emergence of this virus 40 years later was detected in the same region (2000 and 2004) as well as in two other administrative areas (Departments) bordering the Mediterranean Sea, Var in 2003 and Pyrénées Orientales in 2006 [46] (Figure 2). After nine years of apparent absence, in 2015, a large outbreak involving horses occurred in Camargue. Forty nine horses were infected by WNV with 41 exhibiting signs of neuroinvasive disease [47]. Human WNV cases had been notified on rare occasions during the years 2000, with only seven autochthonous human cases of WNF reported in the Var Department during 2003. In 2015, 2017 and 2018, WNV re-emerged in humans with one, two and sixteen urban human cases, respectively, reported as autochthonously transmitted [20]. The re-emergence of WNV in France is therefore cyclical but unpredictable, for the moment, mostly limited to a restricted area, the Camargue region.

The dynamic spread of USUV has been different. This virus was first isolated in 2015 from a resident blackbird (*Turdus merula*) in Haut-Rhin department, near the French-German border. It was also isolated in *Culex pipiens* mosquito pools trapped in 2015 in Camargue [48,49]. Phylogenetic analysis revealed three different lineages (Africa 2 and 3 lineages circulating simultaneous within mosquitoes and Africa 2 and Europe 3 lineages found in blackbirds) indicating three independent introduction events. After this first detection, large USUV outbreaks, affecting avifauna, were recorded in France in 2016 and 2017 with new departments affected every year (Figure 3). In contrast to outbreaks of WNV, USUV spread quickly across the country. In 2018, new USUV outbreaks have been reported throughout large parts of France. The zoonotic risk of this virus is lower than WNV but this is being constantly revised in the light of new detections of this virus in humans. Indeed, a retrospective flavivirus molecular survey on cerebrospinal fluid samples collected in France in 2016 highlighted that one undiagnosed patient, with a facial paralysis, was positive for USUV RNA [50]. The phylogenetic analysis of the USUV genomic sequence revealed that infection was due to the Africa 2 lineage. 

*Culex* spp. mosquitoes are accepted as the primary global transmission vector of WNV and USUV. However, *Ae. albopictus*, an aggressive day-biting insect, is able to transmit DENV and CHIKV and has been established in mainland France since 2004, spreading extensively in the country to at least forty-two departments by 2017 [51]. It can also be considered an important bridge vector for WNV [52]. The emergence of DENV, CHIKV and ZIKV are considered a threat due to the repeated introduction of these viruses by infected travelers returning from endemic/epidemic regions. Since 2010, occasionally imported and sometimes autochthonous cases of all three viruses were recorded in France. For example, during the summers of 2013 and 2014, a total of four autochthonous cases of dengue fever in the Provence-Alpes-Côte d’Azur region [53] and an outbreak of 11 autochthonous cases of chikungunya fever in the Languedoc-Roussillon region were reported [54]. 

The goal of surveillance in France is to rapidly detect arbovirus infection and prevent its onward spread. The WNV surveillance is multidisciplinary with human, equine, bird and entomological components. The surveillance design considered in France is passive and based on surveillance of neurological cases in human (conducted by Santé Publique France, or public health France) and horses (supervised by the Ministry of Agriculture), which is reinforced in Departments from the Mediterranean area during the vector circulation period between the 1st June until the end of October [47]. Equine surveillance is supported by veterinary practitioners and by the RESPE (Réseau d’Épidémio-Surveillance en Pathologie Équine), a network of sentinel and voluntary veterinarians distributed throughout France. Human surveillance relies on the National Reference centre (NRC) for arboviruses (IRBA-Armed Forces Biomedical Research Institute) that surveys during the same period for any hospitalized patient with neurological signs. Laboratory diagnoses are carried out for WNV but also against USUV and Toscana viruses. As soon as a positive case is detected (either a human case confirmed by the NRC for arboviruses or animal cases detected by the National Reference Center on West Nile virus, ANSES), an active equine and human prospective and/or retrospective surveillance is implemented.

In addition, there is monitoring of abnormal mortalities in wild bird, especially priority species such as crows (*Corvus corone*), magpies (*Pica pica*), blackbirds (*Turdus merula*) and sparrows (*Passer domesticus*), as well as nocturnal or diurnal raptors. This is coordinated through a network for the epidemiological surveillance of wildlife diseases (SAGIR network) set up by the National Hunting and Wildlife Agency (Office National de la Chasse et de la Faune sauvage [ONCFS]) and the National Federation of Hunters (Fédération Nationale des Chasseurs [FNC]). Finally an entomological surveillance is delivered by the Interdepartmental Alliance for Mosquito Control on the Mediterranean Coast (Entente interdépartementale de Démoustication [EID] Méditerranée) when the viral circulation is confirmed in an affected area. WNV surveillance, prevention and control activities are described in the national guidelines [55]. The blood safety measures described in this document are in line with the EU directive and require a deferral of all blood donations from areas with ongoing transmission of WNV to humans (Directive 2004/33/EC). No specific blood controls are required for USUV. 

DENV, CHIKV and ZIKV detections rely on human and entomological surveillance. Since 2006, a national plan “anti-dissemination of chikungunya and dengue in mainland France” has been implemented every year during the *Aedes* spp. mosquito breeding period [56]. After the outbreak of ZIKV in South America, this virus was included in the plan (Instruction 2016). At the regional level, enhanced surveillance is implemented in the districts where the mosquito is established from 1st May to 30th November [54]. All clinically-suspected imported cases must be reported to the local health authority. The NRC for arboviruses confirms the suspected case(s), which must be notified to the appropriate health authority. Upon receipt of a suspicion or confirmation statement of a human case, epidemiological investigations with door-to-door surveillance for acute febrile illness, in each location visited by the patient, and vector control actions are developed to limit the risk of indigenous circulation of DENV and CHIKV. The French advisory group (FAG) for safety of substances of human origin is activated as soon as confirmed cluster of CHIKV and DENV autochthonous cases are reported in a French Metropolitan department. The measures available in this case are listed in a guideline to prevent transfusion-transmission for these arboviruses.

### 2.4. Italy

West Nile virus is considered endemic in Italy with repeated emergence during the summer months. This has been heightened in 2018 with the sudden increase in human cases compared to previous years. The Istituto Zooprofilattico Sperimentale Abruzzo e Molise has, in collaboration with the National Health Institute (ISS), the responsibility for data collection and analysis on West Nile disease (WND) and surveillance activities conducted within Italy. The data observed in Italy concerning the number of human cases of WNF (Figure 4) are not significantly correlated with those observed in the EU (Kendall correlation coefficient, tau = 0.6190, *p* = 0.06905). Nonetheless, compared to the overall picture within the EU, the number of cases observed in Italy in 2018 are higher than those observed in any of the previous years.

The surveillance performed in Italy for WNV includes the testing of pools of mosquitos collected using a range of traps. Entomological sampling begins in March with the purpose of detecting WNV in areas before transmission to humans occurs (recently reviewed in [57]). Furthermore, birds belonging to three species, magpie, hooded crow (*Corvus cornix*), and the Eurasian jay (*Garrulus glandarius*) are targeted, with the objective of sampling at least 100 subjects per province in endemic areas, plus a passive collection of dead wild birds of any other species. The surveillance activities are planned at the national level, and the local veterinary services manage their practical execution. Figure 5 shows the collated data for WNV detection within mosquito pools (A), monthly numbers of WNV positive target bird species (B) and the numbers of WNV positive birds collected by passive surveillance (C) between January 2012 and July 2018. A preliminary analysis of this data suggests that the monthly numbers of positive mosquito pools were significantly correlated to the monthly numbers of human cases of WNF (Kendall correlation coefficient, tau = 0.6466, *p* = 5.6 × 10^−11^). The monthly numbers of positive mosquito pools were significantly correlated to the monthly numbers of target birds positive to WNF (Kendall correlation coefficient, tau = 0.6983, *p* = 4.3 × 10^−13^) and the monthly positivity in mosquitos was also correlated to the monthly positivity in wild birds (Kendall correlation coefficient, tau = 0.5548, *p* = 1.5 × 10^−8^). A conclusion of this analysis suggests that the variations in the incidence of infection in mosquitos led to similar variations in the incidence in both the vertebrate reservoirs of the infection (target birds) and in the accidental hosts (human beings and the wild bird general population).

Possible factors responsible for the high incidence of WNF in 2018 include the temperature and rainfall. Comparison of the data on the last month (July 2018), the last season (spring 2018) and the last year (2017) to the entire period on record (years 1800-2018) [58] suggested that neither of the two factors might be sufficient to explain the increase of WND incidence across all susceptible species observed in 2018 in Italy.

### 2.5. Israel

Due to its geographic location and Middle-Eastern climate, Israel has always been endemic for mosquito-borne diseases. Currently, the major mosquito-borne diseases circulating in Israel are WNF and the bovine disease, three-day sickness, or bovine ephemeral fever, caused by the bovine ephemeral fever virus (BEFV) [59]. Other recently diagnosed human cases of malaria, dengue fever, and Zika infection were reported, but only among travelers returning from endemic regions for these diseases [60,61,62]. Usutu virus was detected in mosquitoes, but no avian or human infections have been reported [63]. WNF, on the other hand, was first reported in Israel in the 1950s [64] with occasional diagnosis in human patients and sporadic isolation from mosquitoes [65]. A major outbreak took place during 1997-2000, affecting birds, horses and humans [66,67]. However, identification of WNV in horses and virus isolation from birds and horses was not performed between 2003 and 2018. Sporadic cases of WNF increased during an outbreak between July and August of 2018, in which until 16th of August, 38 human patients were hospitalized with medium-to-severe disease and confirmed as infection with WNV by the Israeli MOH Division of Epidemiology (Figure 6). During this outbreak, the virus was identified by qPCR in 12 different mosquito traps, two horses, one donkey and 9 birds, which were identified as positive for WNV [68]. The recent outbreak in birds, equines and humans after several years of sporadic cases, highlights the importance of constant surveillance for WNV in warmer climatic regions and in densely-populated areas, such as central and northern Israel. This requires combined effort from environmental, veterinary and health professionals at the regional and national level. Genotyping of Israeli WNV samples since the end of the 1990’s showed that the majority of the identified viruses belong to lineage 1, divided into the Eastern European and Mediterranean clades of cluster 2, and cluster 4. A smaller number of identified cases were of lineage 2 [65].

The WNV outbreak during 2000 caused significant economic effects to goose farms in Israel, which necessitated the development of an autologous vaccine to combat the infection [69]. Wild birds are affected by WNV infection, with the virus successfully identified and isolated from several species. However, the accurate extent of damage to wild birds in Israel from WNV is not known. Israeli horse farms are under a constant threat of WNV infection, with an increased seropositive rate, reaching as high as 85% in examined farms [66,70]. On-going surveillance for WNV and other mosquito-borne pathogens is performed by the Israeli Ministry of Environmental Protection (MEP), together with the Ministry of Health (MOH), involving the National Parks Authority (NPA) rangers collecting samples, the MOH entomology laboratory identifying each catch, and the MOH virology laboratory testing for the presence of the pathogens [65]. Birds and equids exhibiting neurological signs are tested by the Kimron Veterinary Institute (KVI). The need to rapidly diagnose WNV infection in birds, equids and humans led to development of improved diagnostic tests, together with strengthened collaboration between the parties involved in surveillance and diagnosis, namely, MEP, MOH, and KVI. 

Bovine ephemeral fever (BEF) was first described in Israel in 1931 with occasional outbreaks every few years [71] and is caused by infection with BEFV. Infections in cattle, as well as seropositive responses detected by the virus neutralization test, are confirmed almost every year by the Kimron Veterinary Institute (Erster, unpublished data). This suggests that although outbreaks occur sporadically, the virus is endemic to Israel. The occasional outbreaks of BEF affects the dairy industry, mainly due to decreased milk production, and increased abortion rate in infected herds. Assessment of the economic impact of the 1999 BEF outbreak was estimated at an average US$112 cost per non-lactating cow and US$280 per lactating cow [72].

The reported study included only 8 farms, each with an average of 250 lactating cows, with approximately 40% morbidity in each herd. Therefore, a gross estimation of the effect of a single outbreak was US$28,000 per herd for lactating cows, excluding non-lactating cows and high milk somatic cell count damage [73]. Bovine ephemeral fever is monitored through blood and mosquito examinations by the KVI. Since 2005, the gene encoding the virus surface protein, glycoprotein G was routinely sequenced, demonstrating that the virus circulating in Israel is, not surprisingly, closely related to the one circulating in the neighbouring countries Egypt and Turkey [74]. A live attenuated vaccine based on an Australian BEFV strain was used in Israel, but its efficacy in conferring immunity against the local BEFV strain was not determined (Erster, personal communication).

Located on a major bird migration route, the geographic location of Israel, combined with a relatively warm climate and the dense human population in the central and northern parts of the country, are all factors that promote the rapid spread of introduced or re-emerging arboviruses. Recent assessment of the possible incursion of exotic arboviruses concluded that the environmental conditions, including the presence of an established vector, could permit potential autochthonous transmission of DENV and CHIKV [75]. However, there have been no reports of transmission of these viruses in Israel. Rift valley fever (RVF), transmitted by mosquitos, has never been reported in Israel, but its re-occurrence in Egypt [76] suggests that its incursion into Israel in the future is a realistic possibility.

### 2.6. Germany 

Surveillance for emerging mosquito-borne disease is based on country-wide collection of mosquito populations [77] and wild birds [78], followed by testing for certain viruses. This approach has led to the detection of USUV, SINV and Batai virus (BATV) in certain mosquito species [79]. A SINV was also found in a hooded crow in Berlin [80]. Usutu virus is now considered endemic following a relatively recent introduction [81] whereas, the detection of SINV and BATV could represent endemic viruses or recurrent incursions from neighbouring countries. A recent development in 2018 has been the detection of WNV in captive great grey owls (*Strix nebulosa*) in the Federal States of Saxony-Anhalt and Bavaria, and wild goshawks (*Accipiter gentilis*) in Saxony-Anhalt. Moreover, two WNV cases were found in dead blackbirds (*Turdus merula*) in the capital, Berlin, and the State of Mecklenburg-Western Pommerania as well as WNF in horses (1 fatal, 1 survived). One human infection with WNV was eventually reported in a pathologist who developed a febrile illness after dissecting a great grey owl in Poing, near Munich, Bavaria [82]. These cases indicate a significant spread of the virus beyond previously affected areas. All cases were located in the Eastern and South-eastern regions of Germany and caused by infection with WNV lineage 2 strains. Preliminary phylogenetic analysis indicates a single introduction as early as 2016 from the Czech Republic and thereafter undetected circulation over a distance of up to 900 kilometres (from Poing, Bavaria to Rostock in Mecklenburg-Western Pommerania). Furthermore, a coding point mutation was found in the NS5 gene, but it is unclear whether this had any effect on the virulence and spread of the circulating WNV strains.

## 3. Discussion

A recent study by the European Centre for Disease prevention and Control (ECDC), the transnational body for Europe that monitors and provides advice on human infections, identified vector-borne diseases as one of the eight major infectious threats for the EU, in “competition” for example, with antibiotic resistance in bacteria or with pandemic influenza [83]. This has been borne out by the introduction and establishment of both WNV and USUV throughout much of Europe, incursions that are continuing. Climate change may be influencing this although the speed at which both viruses are spreading suggests more direct human involvement such as translocation along road networks, or movement by infected birds, may play a more prominent role. This has resulted in seasonal outbreaks of WNF in countries across Europe for which there is no effective control measure such as a human vaccine. Even the availability of an equine vaccine for WNV has not proven a solution as uptake has been low, perhaps due to the sporadic nature of outbreaks and the relatively high cost of vaccine for owners.

All European countries face the introduction of mosquito-borne viruses and malaria from travelers returning from tropical and sub-tropical regions. This trend will continue as the volume of air travel increases and mosquito-borne viruses continue to cause sudden outbreaks throughout Africa, Asia and South America [84]. Until relatively recently, Europe lacked large urban populations of anthropophilic mosquitoes such as *Ae. aegypti* and *Cx. quinquefasciatus* that are found in tropical and sub-tropical regions. The introduction, establishment and spread of invasive *Aedine* mosquito species, particularly *Ae. albopictus* has resulted in autochthonous transmission of both DENV and CHIKV in southern Europe. Control efforts are able to suppress local populations but eradication of *Ae. albopictus* is unlikely and the species is likely to continue to expand into new areas of the continent [85,86]. In this field ‘citizen science’ can play a role in monitoring the presence or absence of mosquito species and mapping the spread of invasive mosquito species [5,87,88].

Surveillance for mosquito-borne viruses has been adopted as the most effective means of predicting the emergence of disease before human populations are affected. By targeting mosquito populations and sentinel species such as wild birds and horses, the aim has been to identify areas where virus circulation is occurring before human infections result. This requires considerable effort to process and test samples, coordination between different organisations dealing with public and veterinary health, and the rapid dissemination of information when virus is detected to prevent mosquito transmission and secure blood supplies. Development of high-throughput testing and multiplex molecular and serological testing has enabled this to be achieved in the timeframes needed to be effective [89,90]. All the countries surveyed in this review undergo a degree of surveillance for mosquito-borne diseases although this varied in proportion to the threat of experiencing infections. A recent review has focused on national and European-wide surveillance for WNV [20]. Data sharing through bodies such as the ECDC enables countries to re-assess the risk of disease spread and consider surveillance options in light of developments across the continent. 

Most European countries have been affected by USUV although both the British Isles and Scandinavia remain free of the disease. Water bodies such as the English Channel and the Baltic Sea, respectively, may prove to be barriers to transmission if mediated by mosquito populations. However, if bird movement and migration are key players in virus translocation then it is likely that both regions will be visited by this virus. It is possible that cryptic spread has already occurred but USUV has not established in local populations of the main mosquito vector, *Cx. pipiens*. Indigenous *Cx. pipiens* in The Netherlands appear particularly susceptible to infection with USUV and capable of transmission [91], whereas UK populations appear to be less able to transmit virus [26]. Whatever the reason, continued surveillance is necessary for both USUV and WNV.

The trends in mosquito-borne virus introduction and spread are likely to continue in future years. The warm summer temperature experienced during 2018 may have been responsible for the earlier emergence of WNV than in previous years [92], the increased numbers of human cases and its emergence in Germany. However, this has not been conclusively demonstrated but the identification of environmental predictors of virus emergence could be very helpful addition in targeting and triggering seasonal control and prevention activities.

## Figures and Tables

**Figure 1 ijerph-15-02775-f001:**
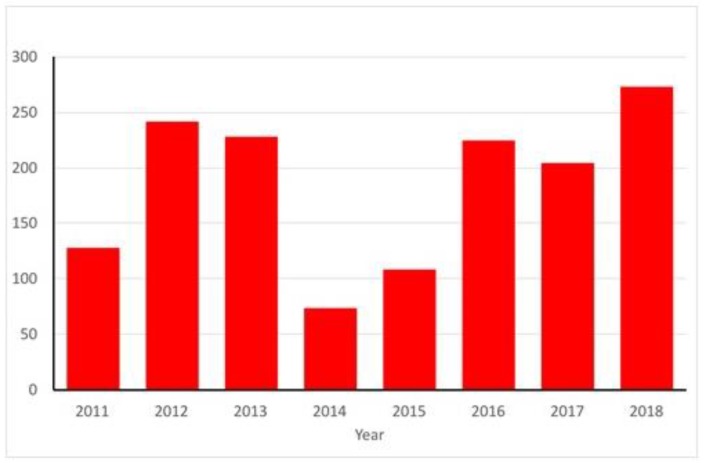
Annual totals for human cases of WNF reported in the European Union. Note that total cases for 2018 are those recorded to 16 August 2018.

**Figure 2 ijerph-15-02775-f002:**
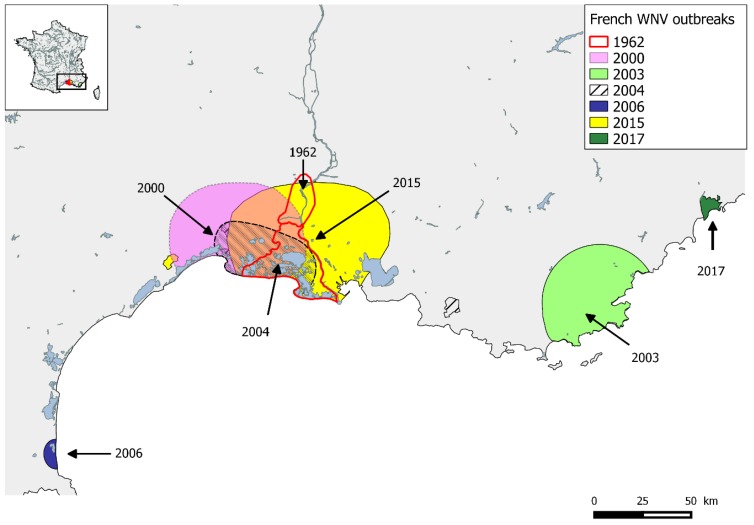
Map of southern France showing locations and years with West Nile virus outbreaks.

**Figure 3 ijerph-15-02775-f003:**
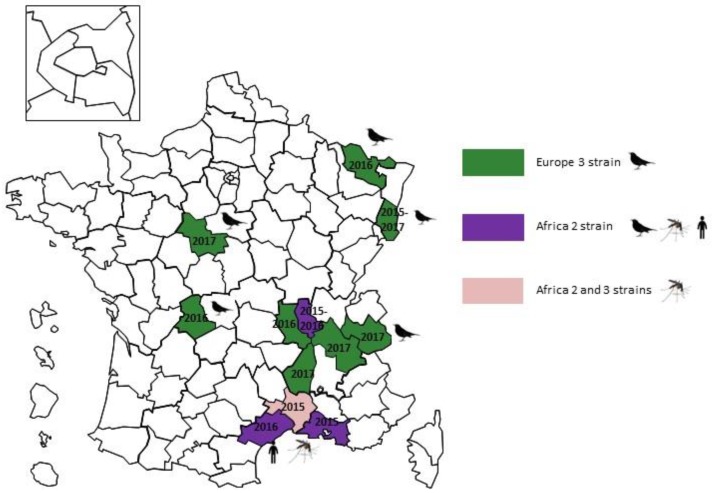
Reports of Usutu virus cases in wild birds in France.

**Figure 4 ijerph-15-02775-f004:**
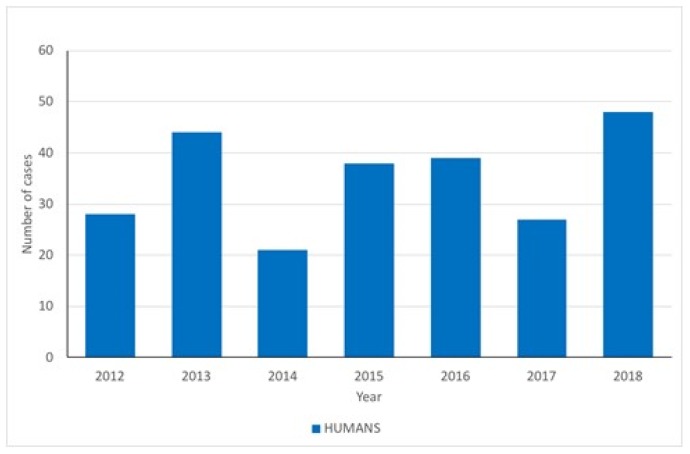
Annual numbers of human cases of WNF in Italy. Note that total cases for 2018 are those recorded to 16 August 2018.

**Figure 5 ijerph-15-02775-f005:**
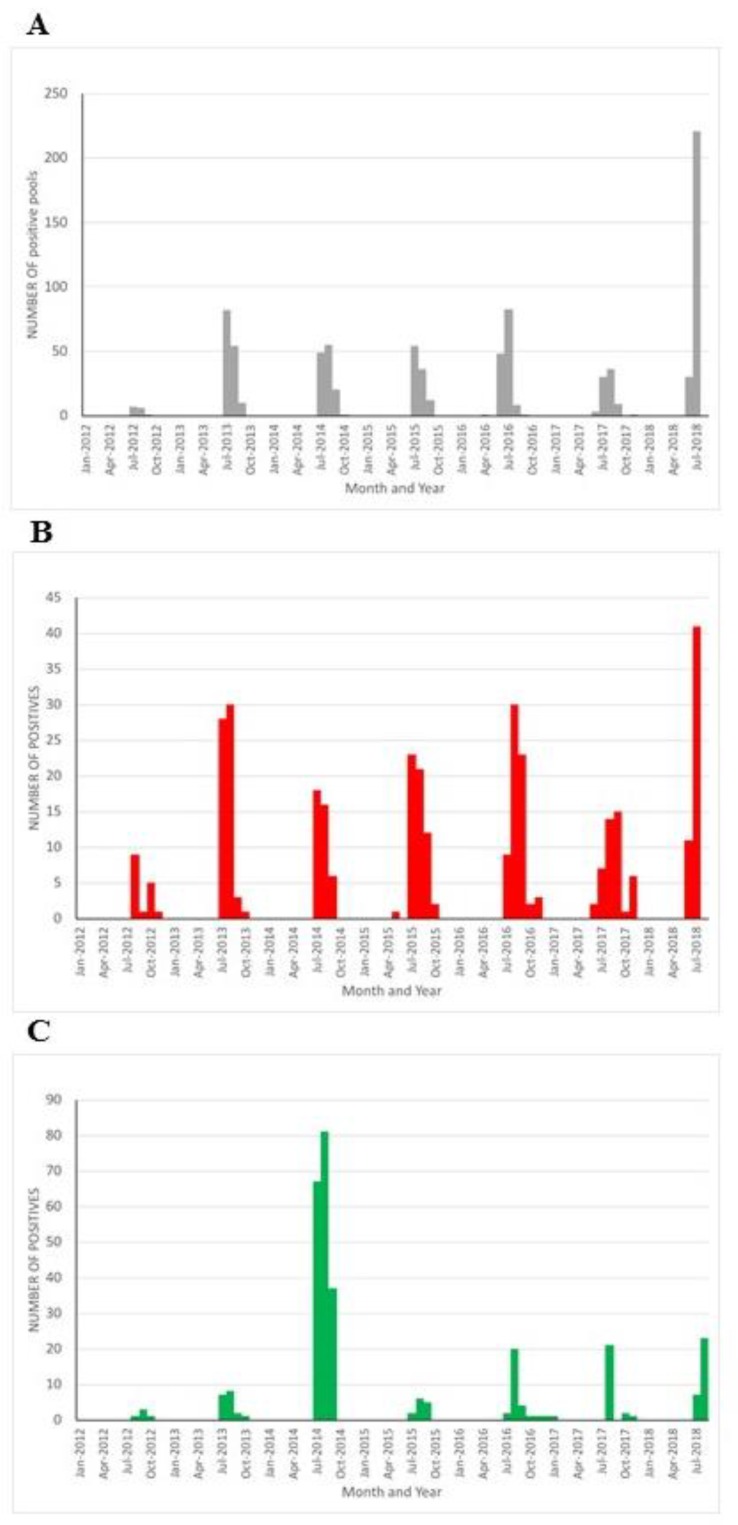
Surveillance effort for West Nile virus in Italy (2012 to July 2018). (**A**) Number of WNV positive mosquito pools reported. (**B**) Number of WNV positive target birds reported. (**C**) Total number of WNV positive wild birds collected during passive surveillance.

**Figure 6 ijerph-15-02775-f006:**
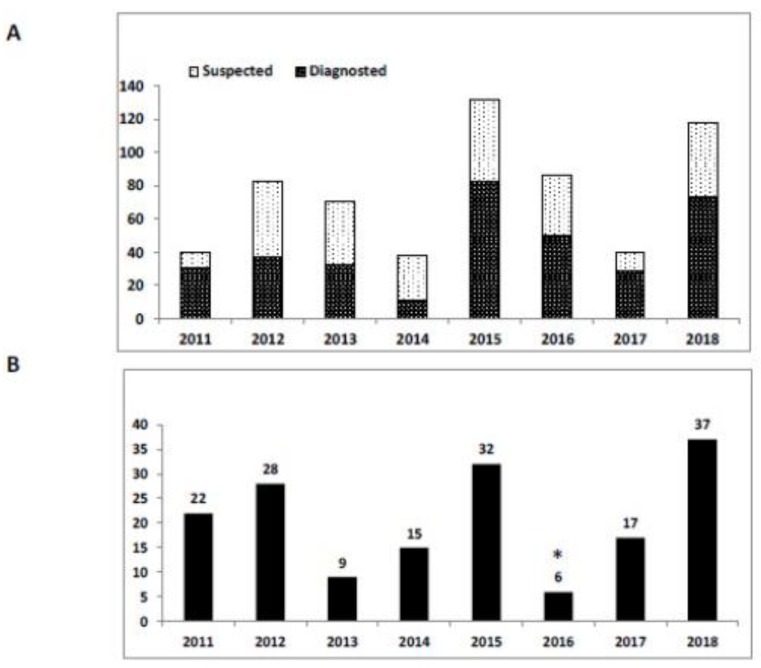
Human case of WNF and WNV positive mosquito pools in Israel during 2011 to 2018. (**A**) Number of human cases in each year. Suspected (pale bars), patients whose preliminary laboratory tests indicated possible WNV infection. Diagnosed (dark bars), patients whose WNV infection was unambiguously confirmed. (**B**) Number of locations from which mosquito pools identified as positive for WNV by PCR. * indicates that surveillance during 2016 was sporadic and underestimates the total number of infected locations.

**Table 1 ijerph-15-02775-t001:** Invasive mosquito species detected in Europe.

Latin Name	Common Name	Origin
*Aedes albopictus*	Asian tiger mosquito	Southeast Asia
*Aedes aegypti*	Yellow fever mosquito	Tropical and subtropical regions
*Aedes japonicus*	Asian bush mosquito	Eastern Asia
*Aedes atropalpus*	American rock pool mosquito	North America
*Aedes koreicus*	None	Korea, Japan, northeast China

**Table 2 ijerph-15-02775-t002:** Summary details of viruses discussed in this review.

Virus Family	Genome Structure	Genus	Species
*Flaviviridae*	Positive strand RNA	*Flavivirus*	*Dengue virus* *Japanese encephalitis virus*
			*Usutu virus*
			*West Nile virus*
			*Zika virus*
*Peribunyaviridae*	Segmented negative strand RNA	*Orthobunyavirus*	*Batai orthobunyavirus*
			*Inkoo virus*
*Phenuiviridae*	Segmented negative strand RNA	*Phlebovirus*	*Rift Valley fever virus*
*Rhabdoviridae*	Non-segmented negative strand RNA	*Vesiculovirus*	*Bovine ephemeral fever virus*
*Togaviridae*	Positive strand RNA	*Alphavirus*	*Sindbis virus* *Chikungunya virus*

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
