# Peer review of "Emerging Mosquito-Borne Threats and the Response from European and Eastern Mediterranean Countries"

_ijerph, 2018, doi:10.3390/ijerph15122775_

Round 1
Reviewer 1 Report
I have reviewed the manuscript titled “Emerging mosquito-borne threats and the European Response” by Johnson et al. The paper presents an overview of the current and future mosquito-borne disease threats and how they are being addressed by local authorities.
Publications of this nature are useful in that they provide an overview of mosquito-related pest and public health threats that stimulate debate and can be used by local authorities to guide management of mosquitoes and the pathogens they transmit.
I feel that the manuscript could be greatly improved by providing greater focus on the differences/similarities between each of the countries profile din the paper. While I appreciate that differences exist with regard to surveillance, it would be useful to provide a table summarising how each country has structured their human, animal, mosquito surveillance programs. As a consequence, an improvement to the manuscript would be to focus on how better surveillance can be incorporated across boarders. A real benefit of this paper would be the strengthen discussion around what is needed for future mosquito-borne disease surveillance across the whole region.
Chikungunya isn’t included as one of the mosquito-borne pathogens in Table 2, given it is discussed later, it would be worthwhile including.
Throughout the manuscript (inc references), scientific names have not been italicised.
There is inconsistency in the abbreviations of pathogens, some are abbreviated and others not. Review for consistency.
It would be useful to mention the role of citizen science in mosquito surveillance across Europe moving forward, there is a number of initiatives that have been undertaken in recent years (and associated papers have been cited here in manuscript) and supporting publications could be cited. It would be a small contribution but worthy of mention.
Minor points:
Line20. Remove duplicate “Abstract”
Line 23. While increased surveillance has added to detection of mosquitoes/mosquito-borne pathogens, surveillance itself is not contributing to the introduction of these pathogens. A rewording is required here.
Figure 1. Remove title from chart, figure caption sufficient
Line 153. Could the authors clarify what “Var department” is?
Line 404. Remove "2. Materials and Methods"
Author Response
Please find attached our responses (in bold) to the reviewers comments. The associated draft contains track changes to indicate where the manuscript has been edited.
Reviewer 1
I have reviewed the manuscript titled “Emerging mosquito-borne threats and the European Response” by Johnson et al. The paper presents an overview of the current and future mosquito-borne disease threats and how they are being addressed by local authorities.
Publications of this nature are useful in that they provide an overview of mosquito-related pest and public health threats that stimulate debate and can be used by local authorities to guide management of mosquitoes and the pathogens they transmit.
I feel that the manuscript could be greatly improved by providing greater focus on the differences/similarities between each of the countries profile din the paper. While I appreciate that differences exist with regard to surveillance, it would be useful to provide a table summarising how each country has structured their human, animal, mosquito surveillance programs. As a consequence, an improvement to the manuscript would be to focus on how better surveillance can be incorporated across borders. A real benefit of this paper would be the strengthen discussion around what is needed for future mosquito-borne disease surveillance across the whole region.
This theme has been developed in the Discussion section including reference to Gossner et al., where a table outlining different approaches has already been presented. We feel it would be inappropriate to repeat this in the current review.
Chikungunya isn’t included as one of the mosquito-borne pathogens in Table 2, given it is discussed later, it would be worthwhile including.
Chikungunya virus has been added to Table 2.
Throughout the manuscript (inc references), scientific names have not been italicised.
These have been italicised.
There is inconsistency in the abbreviations of pathogens, some are abbreviated and others not. Review for consistency.
The review has been revised for consistency of abbreviations.
It would be useful to mention the role of citizen science in mosquito surveillance across Europe moving forward, there is a number of initiatives that have been undertaken in recent years (and associated papers have been cited here in manuscript) and supporting publications could be cited. It would be a small contribution but worthy of mention.
Minor points:
Line20. Remove duplicate “Abstract”
Deleted
Line 23. While increased surveillance has added to detection of mosquitoes/mosquito-borne pathogens, surveillance itself is not contributing to the introduction of these pathogens. A rewording is required here.
The text has been revised.
Figure 1. Remove title from chart, figure caption sufficient
The titles have been removed from Figures 1, 4, 5 & 6.
Line 153. Could the authors clarify what “Var department” is?
A clarification in the text has been made.
Line 404. Remove "2. Materials and Methods"
Deleted.
Reviewer 2 Report
Line 55: italicize scientific names.
Lines 58, 61, 66: See comment above.
Line 72: Israel is not a European country.
Lines 80, 81, 85, 96, & 91 (and continue throughout the manuscript): italics for scientific names! I am tempted to reject based on this alone!
Line 637: Is it permissible to cite a submitted manuscript? Shouldn't this be cited as unpublished data?
I got tired of calling attention to unitalicized scientific names, both in the test an in the references. Clean up the manuscript. The review of literature is fine but I spent time looking for mistakes rather than being educated.
Author Response
Please find attached our responses (in bold) to the reviewers comments. The associated draft contains track changes to indicate where the manuscript has been edited.
Reviewer 2
Line 55: italicize scientific names.
Corrected
Lines 58, 61, 66: See comment above.
Line 72: Israel is not a European country.
Agree and the title has been modified to reflect this. However, ECDC reports data on the country as a ‘neighbouring country’ of the EU.
Lines 80, 81, 85, 96, & 91 (and continue throughout the manuscript): italics for scientific names! I am tempted to reject based on this alone!
Corrected
Line 637: Is it permissible to cite a submitted manuscript? Shouldn't this be cited as unpublished data?
The reference has been changed to unpublished data.
I got tired of calling attention to unitalicized scientific names, both in the test an in the references. Clean up the manuscript. The review of literature is fine but I spent time looking for mistakes rather than being educated.
Reviewer 3 Report
The review by Johnson et al. discusses emerging mosquito-borne threats in Europe. Overall
the review is well written and details provided were useful and kept close to the theme. The inclusion of Isreal to a review was curious, particularly based on the title but not a big deal. outside of small changes, including with hyphenating terms e.g. West Nile Fever (WNF) the authors should stick to the hyphenation throughout and not go back and forth. The review should be accepted.
Author Response
Please find attached our responses (in bold) to the reviewers comments. The associated draft contains track changes to indicate where the manuscript has been edited.
The review by Johnson et al. discusses emerging mosquito-borne threats in Europe. Overall the review is well written and details provided were useful and kept close to the theme. The inclusion of Isreal to a review was curious, particularly based on the title but not a big deal. outside of small changes, including with hyphenating terms e.g. West Nile Fever (WNF) the authors should stick to the hyphenation throughout and not go back and forth. The review should be accepted.
The abbreviations have been checked and the document revised .
Round 2
Reviewer 3 Report
All comments have been sufficiently addressed